# Gender and Autoimmune Liver Diseases: Relevant Aspects in Clinical Practice

**DOI:** 10.3390/jpm12060925

**Published:** 2022-06-02

**Authors:** Federica Invernizzi, Marta Cilla, Silvia Trapani, Maria Guarino, Valentina Cossiga, Martina Gambato, Maria Cristina Morelli, Filomena Morisco, Patrizia Burra, Annarosa Floreani

**Affiliations:** 1Center for Liver Disease, Division of Internal Medicine and Hepatology, IRCCS Ospedale San Raffaele, 20132 Milan, Italy; cilla.marta@hsr.it; 2Italian National Transplant Center/National Institute of Health, 00162 Rome, Italy; silvia.trapani@iss.it; 3Gastroenterology and Hepatology Unit, Department of Clinical Medicine and Surgery, University of Naples “Federico II”, 80138 Naples, Italy; maria.guarino86@gmail.com (M.G.); valentina.cossiga@gmail.com (V.C.); filomena.morisco@unina.it (F.M.); 4Multivisceral Transplant Unit-Gastroenterology, Department of Surgical Oncological and Gastroenterological Sciences, University Hospital of Padova, 35128 Padova, Italy; martina.gambato@gmail.com (M.G.); burra@unipd.it (P.B.); 5Internal Medicine Unit for the Treatment of Severe Organ Failure, IRCCS Azienda Ospedaliero-Universitaria di Bologna, 40138 Bologna, Italy; mariacristina.morelli@aosp.bo.it; 6IRCCS Negrar, 37024 Verona, Italy; annarosa.floreani@unipd.it; 7University of Padova, 35131 Padova, Italy

**Keywords:** gender, autoimmune liver diseases, autoimmune hepatitis primary biliary cirrhosis, primary sclerosing cholangitis, overlap syndromes, liver transplant

## Abstract

Autoimmune liver diseases (AILDs) include autoimmune hepatitis, primary biliary cholangitis and primary sclerosing cholangitis. The etiologies of AILD are not well understood but appear to involve a combination of genetic and environmental factors. AILDs commonly affect young individuals and are characterized by a highly variable clinical course. These diseases significantly influence quality of life and can progress toward liver decompensation or the onset of hepatocellular or cholangiocarcinoma; a significant number of patients eventually progress to end-stage liver disease, requiring liver transplantation. In this review, we focus on the sex characteristics and peculiarities of AILD patients and highlight the relevance of a sex-specific analysis in future studies. Understanding the sex differences underlying AILD immune dysregulation may be critical for developing more effective treatments.

## 1. Introduction

Autoimmune hepatitis (AIH), primary biliary cirrhosis (PBC) and primary sclerosing cholangitis (PSC) represent the three major autoimmune liver diseases (AILDs). Furthermore, the spectrum of autoimmune liver disease includes the overlap syndromes (OS) between AIH and PBC or PSC. Overlap syndromes between PBC and PSC have been described in anecdotal cases only. Genetic predisposition, environmental factors and defects in immune regulation underlie the induction and perpetuation of autoimmunity. AIH, PSC and PBC share common pathways of immune-mediated liver injury, involving the hepatic recruitment of CD4+ and CD8+ T cells, which display cytotoxicity against liver or biliary cells, leading to liver fibrosis, cirrhosis and liver failure [1]. An imbalance between effector and regulatory T cells appears to underlie the loss of immune tolerance to self-antigens in many autoimmune diseases with a poorly understood pathway. Several studies have provided evidence of viral or bacterial triggers in AILD etiology, suggesting that autoimmunity may result from immune recognition of microbial peptides that display sequence similarity to autoantigenic peptides, called molecular mimicry [2]. It is also possible that AILD results from the modification of self-antigens by drugs or micro-organisms, making them immunogenic, or from the aberrant exposure of normally sequestered liver antigens to the immune system as a result of liver damage. The observations that particular MHC class II alleles predispose individuals to developing AILD provide a strong argument that antigen presentation to CD4+ T cells is a central event in the pathogenesis [3]. Although the three diseases exhibit similarities in their pathogeneses, they differ in their patterns of liver injury. AIH is characterized by an inflammatory cell infiltrate, mainly composed of cytotoxic T cells and plasma cells, around the portal tracts, which invades and causes progressive destruction of the liver parenchyma, termed interface hepatitis. In contrast, the large intra and extrahepatic bile ducts are targeted in PSC, leading to biliary tree obliteration and resulting in biliary cirrhosis and portal hypertension. In PBC, the small bile ducts are damaged, leading to portal tract destruction and biliary cirrhosis.

To date, available studies have shown a gender role among autoimmune diseases, such as a higher prevalence of AILD in female patients, with the exception of PSC. On the other hand, male sex seems to have a poor prognosis, apart from in AIH, where the mortality is higher in the female than male sex. The aim of this paper is to review the current knowledge regarding the influence of sex on the setting of AILD, underlining the relevance of sex-specific analysis in the prognosis and management of these diseases.

## 2. Autoimmune Hepatitis

AIH affects the female sex more than the male sex across all ethnicities and ages (children, 60–76%; adults, 71–95%) [3,4,5,6,7,8,9,10,11,12,13,14]. The male-to-female ratio in the population with AIH is considered to have changed over time, indicating a relative increase in the number of male patients. In Japan, the male-to-female ratio was 1:7 in 2004 and 1:4 in 2016 [15]. It should also be stressed that cases with “acute” presentation (transaminases higher than 10 times the upper limit and bilirubin higher than 5 mg/mL) are increasing worldwide. In an Italian multicenter study, among 479 patients diagnosed as AIH, 202 (43%) met the criteria for “acute” onset, and no significant differences were observed between the sexes [16].

Factors responsible for gender differences in immune response are illustrated in Figure 1.

*Sexual hormones* influence innate immunity. High levels of estrogen reduce the synthesis of interleukin-1β (IL-1β), IL-6 and tumor necrosis factor (TNF) by macrophages and monocytes. On the other hand, low levels of estrogen increase the same cytokines [17]. However, in cases in which cytotoxic T cells target liver cells, estrogen may shift the onset of AIH to the late reproductive phase. This is the case of AIH with onset after menopausal state. Moreover, depending on the reproductive status, before puberty, B-cell-dependent autoimmunity plays a key role in AIH, with the production of autoantibodies [17]. Indeed, estrogen also plays a role in natural killer (NK) cells; low estrogen levels increase the activity of NK cells, as well as that of dendritic cells. Estrogen treatment (compared to pregnancy levels) was found to decrease TNF and IL-12 production in mature mouse dendritic cells [18]. These changes also explain the beneficial effect of pregnancy for AIH. During pregnancy, high levels of estrogen and progesterone exert a tremendous inhibitory influence on most inflammatory pathways, but females might be prone to disease flare thereafter in the vulnerable postpartum phase.

*Genetic factors* linked to X chromosomes have been extensively studied [19,20,21,22]. During embryonic development, one of the two X chromosomes is randomly inactivated in females. This process is initiated by long non-coding RNA X inactivation and results in a cellular mosaicism, where about one-half of the cells in a given tissue express either the maternal X chromosome or the paternal X chromosome. However, X chromosome inactivation is not complete, with 15 to 23% of genes escaping inactivation, thereby contributing to the emergence of a female-specific heterogeneous population of cells with biallelic expression of some X-linked genes. Although X chromosomes provide clues as to the cause of PBC [19], few studies have been published on AIH.

*Epigenetic factors* were extensively reviewed by Liu et al. [20]. An important issue is the role of X-chromosome-located microRNAs in immunity, which has been hypothesized to contribute to the enhanced immune response of females [21]. The human X chromosome contains 10% of all microRNAs detected to date. According to recent studies, in several mammalian species, including humans and mice, the X chromosome has a higher density of miRNAs, whereas the Y chromosome has only four miRNA sequences (although not experimentally validated) with two shared by both sex chromosomes [22]. Indeed, miRNAs on the X chromosome may influence sex-specific responses. It has been shown that bone-marrow-derived mesenchymal stem-cell-secreted miR-223-containing exosomes prevent liver injury in an autoimmune hepatitis mouse model by suppressing hepatic NLRP3 and caspase-1 [23]. Moreover, modification of miR-223 further improves its therapeutic efficacy against AIH [23]. In general, miR-223 is involved in the pathogenesis of various liver diseases by influencing immune cell differentiation, neutrophil infiltration, macrophage polarization and inflammasome activation by both metabolic and inflammatory signaling pathways [24]. However, a few studies have been conducted on the epigenetics of AIH utilizing miRNAs, but altogether, these studies suggest the possibility that miRNAs could be used as biomarkers for diagnosis and prognosis of AIH [25].

The *microbiome* can influence a number of physiological aspects of the host, including the immune response, and several studies have attempted to unravel the role of the microbiome in the pathogenesis of autoimmune diseases [26,27]. Moreover, the sex difference in this setting is currently not sufficiently explored, representing a gap that needs to be filled.

Neoplastic risk for hepatocellular carcinoma (HCC) in AIH is lower than other types of liver disease [28]. HCC develops in patients with AIH and cirrhosis in 1–9% of cases, and the annual incidence in patients with cirrhosis is 1.1–1.9% [29], with the same proportion in females and males [30]. Besides cirrhosis, other risk factors linked to HCC are older age, increased frequency of relapses, concurrent alcohol consumption and a trend for male sex [28].

The long-term outcome of patients with AIH was evaluated in 238 patient (51 men) at a single center from 1971 to 2005 [31], showing increased survival in males; however, the reason was unclear. Moreover, the age at death in women with liver-related causes compared to non-liver causes was significantly lower (39 years. vs. 70.5 years., *p* = 0.001). Interestingly, HLA a1-B8-DR3 (associated with increased susceptibility to AIH) was more than twice as prevalent in males compared to females. When considering a large cohort of patients including all ages, patients diagnosed under the age of 18 years were found to have a significantly reduced life expectancy [32] and were at high risk of relapses and liver transplantation (LT), with no significant differences observed between the sexes [32].

### 2.1. Management of AIH

There are no differences in response to therapy according to gender. The goals of treatment include (i) the induction of remission, (ii) the maintenance of remission and (iii) the prevention of progression to cirrhosis and its complications. Standard therapy includes prednisone alone or in combination therapy with azathioprine (AZA). Combination therapy with low-dose prednisolone permits a reduction in steroid-related side effects in the induction phase. The open questions include (i) the outcome of therapy, (ii) the possibility of discontinuing treatment and (iii) alternative or new therapies that could enable better control of the disease. The outcomes of therapy include biochemical remission, complete remission, relapse, treatment failure and stabilization. For patients who have experienced treatment failure or failed to achieved remission, second-line therapies with mycophenolate mofetil (MMF), calcioneurin inhibitors (tacrolimus (TAC)) or cyclosporine have been proposed. In patients with “acute” onset of AIH characterized by jaundice and concomitant coagulopathy a rapid referral to a transplant center may be necessary. This is the case for subjects with acute encephalopathy because it has been shown that this condition is associated with high mortality in cases of corticosteroid therapy [33]. In general, patients with “acute” onset and without hepatic encephalopathy respond well to oral or intravenous steroids (1 mg/kd/day) [34]. Discontinuing treatment is still an important problem. Whereas the British literature in the 1980s recommended indefinite treatment, current AASLD guidelines consider discontinuing treatment a possibility during the course of the disease [4].

### 2.2. Drug-Induced Liver Injury (DILI) with Autoimmune Hepatitis Characteristics

This is a particular form of DILI mimicking AIH with autoantibodies. The nomenclature was introduced by Czaja in 2011 [35]. Numerous drugs can induce this condition, including nitrofurantoin, hydralazine, methyldopa, interferons and checkpoint inhibitors. A consensus conference on this important issue was held on 1–3 March, 2022 in Parador de Nerja (Spain) under the auspices of EASL. In a recent prospective study from 2004 through 2011, 88 cases of DILI with AIH features were analyzed by the DILI Network [36]. Female sex was preponderant (100% in 42 cases due to nitrofurantoin, 79% in 28 cases due to minocycline, 100% of 11 cases due to methyldopa and 71% of 7 cases due to hydralazine). Overall, 40% of subjects showed spontaneous improvement in liver tests after discontinuation of the implicated drugs [37]. Analysis of liver biopsies revealed that it is very difficult to distinguish between histologic features that favor DILI and those favoring AIH. Recently, DILI with AIH features was reported with the use of immune-activating agents, such as checkpoint inhibitors [38]. Liver changes usually improve with steroid therapy, but some cases are resistant and associated with bile duct injury [39].

## 3. Primary Biliary Cholangitis

A significant female preponderance is a well-known clinical feature of PBC, whereas differences between sexes in the clinical presentation at PBC diagnosis are not so well-defined. Twelve studies including 51,290 PBC patients worldwide were analyzed to evaluate sex-related differences at PBC diagnosis [40,41,42,43,44,45,46,47,48,49,50]. Seven studies showed that male sex is associated with delayed diagnosis and, consequently, older age at PBC identification [40,43,44,45,47,49,50]. Six studies showed that male patients received PBC diagnosis at more advanced and severe liver disease with cirrhosis and its decompensation events, as well as portal hypertension signs [41,42,44,45,48,51]. Three studies found that male PBC patients presented with worse liver biochemistries [43,44,51]. Finally, six studies showed that fatigue is more associated with female sex at PBC presentation [41,42,44,45,48,51]. Furthermore, Marzioni et al. analyzed data from electronic medical records of patients from 900 general practitioners in Italy, identifying 412 PBC patients and showing that osteoporosis, inflammatory arthritis and other connective tissue diseases were significantly more common in women, whereas inflammatory bowel diseases were significantly more common in men (*p* < 0.01) [52].

The clinical impact of PBC is highly variable, and one of the most important factors contributing to this variability is the response to primary therapy with ursodeoxycholic acid (UDCA) [53]. Several studies have demonstrated that the efficacy of UDCA therapy strongly determines long-term outcomes [53,54,55]. Moreover, many studies have evaluated the role of sex in the response to UDCA therapy, although results are conflicting because of small sample size, retrospectivity and different criteria used to determine response. Eleven studies including 9748 patients with PBC were analyzed [40,48,55,56,57,58,59,60,61,62,63]. All of these studies, except one [59], evaluated the impact of sex on UDCA therapy response. In particular, four of studies showed that no response to UDCA is more frequent in the male than female sex [40,56,57,58]. However, although these studies presented large sample sizes (7677 PBC patients), they evaluated response to UDCA with four different criteria, and this heterogeneity makes the results not comparable. Particularly, Carbone et al. [40] used the UK-PBC cohort to evaluate UDCA response with Paris I criteria and demonstrated that sex is an independent predictor of therapy failure. Instead, Cheung et al. [56] analyzed the largest cohort of PBC patients utilizing the GLOBE score criteria. Finally, Lammert et al. [57] used the Toronto criteria, and Tian et al. [58] utilized a combination of the Barcelona and Paris I criteria. A similar heterogeneity in the evaluation of UDCA response was observed in six, studies showing that sex has no impact on response to therapy [48,55,60,61,62,63]. In 2016, obeticholic acid (OCA) was approved as a second-line therapy in PBC patients with inadequate response or intolerant to UDCA [64]. Four studies evaluated the response to OCA therapy according to POISE criteria [65,66,67,68]. Only D’Amato et al. [68] considered the role of sex in OCA therapy and showed that sex has no impact on inadequate response to OCA.

There is currently little information available regarding the exact magnitude of HCC risk in PBC patients according to sex. A recent meta-analysis evaluating 18 studies examining the incidence of HCC in PBC patients according to sex showed a pooled HCC incidence rate of 9.82 per 1000 person-years (95% CI 5.92–16.28) in men and 3.82 per 1000 person-years (95% CI 2.85–5.11) in women, with moderate-to-high between-study heterogeneity [69]. Additionally, Trivedi et al. showed that HCC incidence was higher in male UDCA non-responders versus responders (HR 4.44, 95% CI 1.29 to 10.20; *p* < 0.001) in a cohort of 4565 PBC patients [70]. Moreover, Harada et al. found that the cumulative incidence of HCC was 6.5% in males and 2.0% in females (*p* < 0.0001) during the 10 years after PBC diagnosis, indicating that male PBC patients had a 3.3-fold higher risk of HCC compared with female PBC patients [71]. Nonetheless, although PBC primarily affects females, the authors postulated that HCC might be more common in male PBC patients because of a lack of estrogen-mediated prevention. In females, the HCC incidence gradually increased according to histological stage, indicating that the terminal stage of PBC, which is a cirrhotic state, may be a risk factor for HCC development in females, whereas males are likely to develop HCC at any stage [71].

The role of sex in the prognosis of PBC patients has been widely evaluated. Seven studies analyzing the presence of ACLD were considered [42,45,48,51,55,56,72]. All of them showed that the presence of ACLD was more frequent in male than in female patients. In particular, Cheung et al. [56] and Marschall et al. [42] demonstrated that male sex had a higher prevalence of portal hypertension and liver decompensation. Moreover, Adejumo et al. [72] showed that male sex had higher risk of jaundice, spontaneous bacterial peritonitis and acute liver failure, whereas female sex had a higher risk of hospitalizations. Furthermore, 11 studies evaluated the role of sex in mortality for PBC patients [42,47,48,50,53,55,56,73,74,75,76]. Eight of these studies demonstrated that males have a higher risk of mortality. In particular, John et al. [55] showed that male sex is a risk factor for death, liver-related mortality and liver decompensation. Lleo et al. [74] demonstrated that male sex was associated to an increased risk of all-cause mortality.

### Primary Sclerosing Cholangitis

PSC is considered an immune-mediated disease with atypical features, including prevalence in men, the absence of disease-specific autoantibodies and poor response to immunosuppression. Regarding gender distribution, PSC affects men prevalently; in a large regional population study from Sweden, the mean crude annual incidence of PSC was 1.22 per 100,000 in the total population aged ≥18 years in the period from 1992 to 2005; among men and women, the incidence was 1.8 and 0.7, respectively. The point prevalence of PSC in the same population was 16 (24 among men and 9 among women) per 100,000, and the proportion of men was 71% [77]. In other nations, proportion of men ranged from 51% in New Zealand [78] to 71% in the USA [79] and in Norway [10]. Similarly, in a recent data collection from the National Rare Diseases Registry (RNMR) and the National Mortality Database (NMD) in Italy, 60% of new PSC diagnoses were in male patients, with a male-to-female ratio of 1.5:1 [80]. Mean age at disease onset was 33 years (SD = 17), and mean age at diagnosis was 37 years. There were no statistically significant differences in age at diagnosis, age at onset and diagnostic delay between male and female patients. In other studies, the median age at PSC onset was generally higher in women than men; a large cohort study from Germany published in 2018 evaluated patients with late disease onset (defined as first diagnosis after 50 years), revealing that the proportion of females was significantly higher in the late-onset group compared with the earlier-onset group (50/183 (27%) vs. 15/32 (47%), *p* = 0.02) [81]. A study population from Sweden was reported to have a time-trend increase in the incidence of large-duct PSC among women but not among men; conversely, the incidence of small-duct PSC increased significantly among men but not among women. Diverging trends were also observed for the incidence of PSC related to IBD, with a significant increase in the incidence of PSC-IBD in women, whereas in men, an increase in PSC without IBD was observed [77].

Patients with PSC are at increased risk of developing several hepatobiliary cancers, mainly cholangiocarcinoma, gallbladder and colon cancer—and hepatocellular carcinoma to a lesser degree. *Cholangiocarcinoma* (CCA) is a model of malignancies occurring in the inflammatory background. Chronic inflammation of the biliary tree induced by PSC promotes oncogenesis and predisposes to development of CCA through DNA damage, cellular proliferation and oxidative stress [82,83]. The annual risk for CCA in PSC is approximately 2%, with a 10- and 30-year cumulative incidence of 6–11% and 20%, respectively, and an increase of 400-fold when compared with the general population [84,85]. Regarding the sex-specific risk of CCA development, female sex seems to be associated with a lower risk of CCA (HR,0.68; *p* < 0.001, respectively), as reported in the data from a large international PSC cohort, which included 7121 patients encompassing >30 years of clinical observation. According to multivariate analysis, advancing age at diagnosis is an independent risk factor of CCA development, whereas female sex and having small-duct disease or CD at the time of PSC diagnosis are protective factors against CCA development [86].

Inflammatory bowel disease occurs in 70–80% of patients with PSC, and PSC seems to confer additional risk of developing *colorectal cancer* (CRC) when compared with the risk in patients with IBD alone [87,88]. CRC can appear in up to 20–30% of PSC-IBD patients, and an annual colonoscopy is recommended [89].

In 2020, a nationwide population-based study from national healthcare registries in England identified incident cases of IBD with and without PSC over ten years; the study showed that patients with PSC-IDB younger than 40 years old had a fourfold higher risk of CRC, whereas there was no difference between groups for patients in which the IBD diagnosis was made in patients older than 60 years. Regarding sex differences, the risk for CRC was significantly lower among women than men (HR 0.46 *p* < 0.001) [90]. The oncologic additive risk of association between PSC and IBD was confirmed in a Spanish multicenter retrospective cohort study. The risk of CRC was increased four- to fivefold in PSC-IBD patients compared to IBD controls, including patients submitted to annual colonoscopic surveillance [91].

Data related to sex differences in PSC clinical presentation and evolution are scarce. From the Italian registry, including 502 PSC patients in population-based data, the survival rate was 92% at 10 years from diagnosis and 82% at 20 years, considering all causes of deaths. The Kaplan–Meier curves show no significant difference between male and female patients with respect to estimated survival times from diagnosis [80]. The International PSC Study Group published a multicenter outcome study in 2017 to describe the natural history of the disease, including 7121 patients across 17 countries and encompassing ≥30 years of clinical observation from 1980 through 2010. The study registered sex-specific variations in clinical phenotype and correlations with liver disease progression and neoplastic complications. Men comprised the majority of the cohort (66%) and were younger than women (average age of 37 years vs. 40 years). Women more commonly exhibited small-duct PSC phenotype, ulcerative colitis (UC) was less common in women than men (48% vs. 61%, respectively *p* < 0.001) and small-duct PSC was characterized by a low-risk phenotype in both sexes (adjusted HR for men, 0.23; *p* < 0.001 and adjusted HR for women, 0.48; *p* = 0.003). Female sex was an independent protective factor against liver progression; in particular, females maintained a significantly higher transplant-free survival than males matched for age and PSC phenotype. Moreover, the lower prevalence of UC in women may partially account for differences in liver disease progression between the sexes.

## 4. Overlap Syndromes and Gender

The term overlap syndrome (OS) describes a subtype of clinical syndromes that share common features relating to AIH and PBC or AIH and PSC. Very rarely, an overlap syndrome between PBC and PSC has been described. However, the term remains controversial, and it is not known whether overlaps are situations occurring simultaneously or represent a different development during the natural course of the disease.

PBC-AIH is the most common form of overlap syndromes [92,93]. A systematic review included 17 studies of PBC-AIH comprising a total of 402 patients [94]. Female gender was present in 87–100% of either retrospective or prospective studies.

AIH-PSC overlap syndrome has been described in both children and adults. In children, the syndrome is particularly important, reported in up to 40% of patients with AIH [95]. Adults diagnosed with AIH-PSC overlap are significantly younger at the time of diagnosis than those with classical PSC (24–27 years vs. 39–46 years, respectively) [96,97,98]. The proportion of adult males with AIH-PSC overlap is 69–81%, which is higher than in AIH [99]. However, the proportion of male gender in AIH-PSC undergoing liver transplantation was slightly lower (49%) than the number reported in classical PSC [100].

### 4.1. Concurrent Autoimmune Disorders in Patients with Autoimmune Liver Diseases: The Effect of Gender

AILDs often coexist with other extrahepatic autoimmune diseases (EHAIDs). Notably, autoimmune thyroid disease and Sjogren’s syndrome (SS) are the most common EHAIDs. The incidence of EHAIDs in patients with AILD is different in AIH, PBC, PSC and OS [101], and there is a lack of data about the effect of gender on the prevalence of EHAIDs in AILD (Table 1).

An Italian study of 327 AIH patients, showed a significant prevalence of EHAIDs (69% had pure AIH and 31% had EHAIDs). The prevalence was higher in females (72% of pure AIH cases were female; 90% with EHAIDs were female); the most frequent association was with AITDs, which were found in 51% of the 101 patients with EHAIDs [102]. In a Danish cohort study of 2745 patients with AIH (71% women), the prevalence of EHAIDs was higher in women than in men. Among AIH patients, 193 (7%) had two or more EHAIDs, with a maximum five [103]. Multiple logistic regression analysis with regard to PBC showed that only female gender was significantly associated with positivity for EHA conditions [104]. Cumali Efe et al. collected data of 1554 patients with PBC diagnosis between 1994 and 2017 in 20 centers from Europe, USA and Canada. A total of 35 different EHAIDs were diagnosed in 440 (28.3%) patients with PBC; among these, 358 (23%) had one associated EHAID, and AITDs were the most common (11%). Patients with EHAIDs were more commonly female (93% vs. 86%, *p* < 0.001) [105]. Another Italian study enrolled 361 PBC patients between 1975 and 2012 (22 males, 339 females); 61% of them had EHAIDs, and 39% had pure PBC. In this study, female patients with EHAIDs accounted for 97% of the sample, whereas male patients accounted for 3%. Female patients with pure PBC accounted for 89% of the sample, whereas male patients with pure PBC accounted for 11% [104]. A recent Chinese retrospective study enrolled 505 patients with PBC (65% with pure PBC, 26% with PBC and SS association and 7% with PBC-AIH). Notably, the proportion of female patients was found to be significantly higher in the PBC-AIH (91%) and the PBC-SS (81%) groups, indicating that female gender was significantly associated with positivity for EHA conditions (*p* < 0.05) [106]. The opposite is true of PSC, where the development of ulcerative colitis was less common in women than men (48% vs. 61%, respectively; *p* < 0.001) (9). Globally, 50–80% of patients with PSC have concomitant inflammatory bowel disease [107]. Besides overlap of these AILDs, the observation of concurrent diverse autoimmune diseases has been reported frequently in patients with AIH and PBC. Cumali Efe et al. conducted a study of 71 AIH/PBC patients (58 female, 13 male); 31 had at least one EHAID (AITDs were the most common), but no differences in sex were described [108].

### 4.2. Impact of Gender on Liver Transplantation for Autoimmune Liver Diseases

To date, a significant number of AILD patients have been reported to eventually progress to end-stage liver disease requiring LT. LT in AILD is indicated when liver failure occurs with complications similar to those for end-stage liver disease caused by other etiologies. With regard to PSC, due to the variability of the course of the disease, timing of transplantation is difficult to predict; the risk of development of malignant disorders in the liver/biliary tract, recurrent bacterial infections in the biliary tract and the impact of inflammatory bowel disease (IBD) are all aspects to consider [109]. In particular, both EASL and AASLD guidelines recommend that patients with biliary dysplasia be considered for transplantation to remove malignant development at an early stage before progression to invasive cholangiocarcinoma. A small group of patients diagnosed with cholangiocarcinoma but with very limited disease can also benefit from transplantation. Selection of patients is crucial, and treatment includes neoadjuvant radiochemotherapy according to protocols [110,111]. An unacceptable quality of life because of severe, treatment-resistant pruritus, severe hepatic encephalopathy or recurrent cholangitis may also merit consideration for transplantation. Fatigue in PBC and other cholestatic liver diseases is often severe and disabling. Cross-sectional studies have shown no evidence of improved fatigue after LT, whereas others demonstrated that fatigue can improve after LT, although only significantly in 50% of cases. Whether this improvement is enough to justify organ allocation in patients with fatigue alone, without liver failure, remains an open issue [111,112]. Finally, among the three AILDs, only AIH presents as acute liver failure and hence qualifies patients for high-urgency (HU) liver transplantation [113]. Sex may affect the severity of autoimmune diseases, the disease course and, consequently, the indication to LT [112,114]. Each of the three liver diseases accounts for 2% to 6% of the indications for LT according to the European Liver Transplant Registry [115]. PSC accounts for about 4–5% of European LTs, with a prevalence that seems to be stable over the years [114]. The proportion of LTs for AIH has also remained stable over the time (2%); in contrast, with regard to PBC, a falling transplant rate was registered in Europe, dropping from 8% (from 1988 to 2001) to 4% (from 2000 to 2009). Reasons for this decline may relate to diagnosis in an earlier stage and more effective treatment [112]. With respect to gender differences, men with PBC are older and seem to have a more severe disease course, a higher incidence of HCC and higher overall mortality compared with female individuals [114]. A ELITA (European Liver Transplantation Association) study including LT patients from 1986 to 2016 found that despite a relative decrease, the absolute number of transplantations for PBC is steady on an annual basis. These patients, predominantly female, are slightly older, have higher MELD scores and are more likely to be male compared to 30 years ago. Males were significantly older than females at the time of transplantation, whereas the MELD score did not differ between the sexes [116]. As regards AIH, male patients present at a younger age and show a higher relapse rate compared with females [114]. Comparing overall mortality and need for LT, men with AIH appear to have better survival compared to female patients. Despite this, the proportion of patients who required LT or died because of liver-related illness was not significantly different [31]. Data related to sex differences in PSC are not particularly thorough. Whereas the literature points toward an almost equal sex distribution and no significant differences in age at diagnosis between male and female patients, information regarding sex differences in disease severity is scarce and contradictory. In an American study based on an international online registry established in 2014 among PSC patients or their caretakers, Kuo et al. compared symptoms, disease progression and treatments of PSC and found, in contrast to previous studies, a higher proportion of female individuals (53%) [117]. In a United Network Organ Sharing study assessing outcome after waitlisting for LT in a set of 8272 adults with PSC, young PSC patients were found to be predominantly male individuals (70%) [118].

### 4.3. Liver Transplantation and Recurrence

In general, LT in this setting has a favorable overall outcome, with current patient and graft survival for all indications in Europe ranging from 64 to 80% at 5 years, depending on the indication (HCC or not), etiology and severity of the disease [51,112,119]. Nevertheless, all three conditions may recur after transplantation and are associated with an increased risk of both acute cellular and chronic ductopenic rejection.

In PSC, recurrent disease (rPSC) affects 10% to 27% of recipients, with an 8.4% graft loss rate due to recurrent disease. An intriguing, well-documented risk factor for rPSC is the link with IBD. Specifically, the absence of inflammation in the intestine, either due to the absence of concurrent IBD or colectomy before or during LT has a protective effect against (rPSC) [112,120], although not all reports concur with this observation.

Males were found to have more graft failure (due to chronic rejection or PSC recurrence) or experience acute rejection, suggesting that age- or sex-related differences can impact outcomes pre- and post-transplantation [118,119,121]. These results were confirmed by a recent study by Berenguer M. et al., which demonstrated that despite no differences in cause of death, post-transplantation (after year 2000) outcome, particularly graft survival, was worse for men than women (10 y graft survival of 56% for men versus 63% for women). Additional factors associated with worse outcomes presented no differences globally in terms of sex and included older recipient and donor age, the presence of CC at LT, the reduced use of grafts and prolonged ischemia time (only for grafts) [114]. Compared to other autoimmune diseases, rPSC is associated with decreased graft survival, and there is no established treatment either before or after LT. Some centers continue to use UDCA for rPSC, as it improves liver biochemistry; nevertheless, it does not improve survival. Given the unmet therapeutic need of PSC patients in general, other modalities to improve bile acid flow and composition are being actively studied.

In autoimmune hepatitis, recurrence affects approximately 25% of liver allografts during the first 5 years after LT and more than 50% after 10 years of follow-up, with 6% graft lost due to recurrence [119,121]. Long-term outcome and prognosis are more favorable in men with AIH compared to women. The reasons for this remain unknown but may reflect either gender alone or the effect of gender on immune responses [31]. Risk factors for recurrence of AIH (rAIH) after OLT have been assessed in several studies, but most remain unvalidated and controversial. Discontinuation of steroid therapy, HLA-DR locus mismatching, elevated IgG before LT and moderate-to-severe inflammation in the explant are reported as significant risk factors for recurrence of AIH, suggesting that recurrence of autoimmune hepatitis may reflect incomplete suppression of disease activity prior to LT [112,120,122]. The treatment of rAIH is empiric and very much depends on the presentation, which can be variable. In asymptomatic disease and with minimal changes in liver biochemistry or histology, minor adjustments with increased immunosuppression may be sufficient to suppress recurrent disease. In more active rAIH, more potent regimens tend to be employed with either an increased dose, recommencement of corticosteroids and/or addition of immunosuppressive agents. Re-transplantation may be required for patients with rAIH who present with liver failure and graft loss [120].

Recurrent PBC is reported in a range from 17% to 42% after LT; however, in contrast to AIH and PSC, graft loss due to recurrent disease is not a major issue in PBC (1.3%) [119]. Sex differences among PBC recurrence are lacking. Several studies have shown that tacrolimus-based immunosuppression is associated with an increased risk of recurrence of PBC, with a reduced time to recurrence compared with ciclosporin. The role of genetic factors has not been investigated thoroughly. The human leukocyte antigen (HLA) profile and HLA donor–recipient mismatch have a controversial association in rPBC [120,122]. Case series have reported that the development of rPBC has little impact on long-term survival or need for re-transplantation. However, this may be related to the small number and short follow-up of patients with rPBC. To date, ursodeoxycholic acid (UDCA) is the only drug accepted for the treatment of patients with PBC and is generally employed after a diagnosis of rPBC has been established. The use of UDCA has been associated with improved liver biochemistry tests in patients with rPBC; however, we lack data documenting a delay in histological progression or improvement in graft and patient survival [120].

### 4.4. De Novo Autoimmune Hepatitis after Liver Transplantation

De novo AIH is a clinical entity resembling AIH that develops in LT recipients transplanted for other liver disorders. It shows characteristics atypical for AIH, including lymphocytic cholangitis, central perivenulitis and other features consistent with T-cell-mediated rejection. It was originally described in children after LT, predominantly in those with biliary atresia, and was subsequently found in a higher prevalence of LT recipients with PBC. The incidence of de novo AIH is variable because multiple descriptions have been used in case series; however, the disease is rare and does not appear to have an impact on long-term survival. Recipients of female grafts and older donors have a higher prevalence of de novo AIH, suggesting that the risk of AIH may be harbored in the allograft. With regard to immunosuppression, patients maintained on tacrolimus or mycophenolate mofetil have a higher risk of developing de novo AIH, whereas LT recipients treated with ciclosporin have a reduced risk [123].

## 5. Conclusions

The mechanisms behind the sex differences observed in autoimmune liver diseases, specifically the female predominance in AIH and PBC; the worse disease course in male PBC; male predominance in PSC patients with a lower risk of UC; and cholangiocarcinoma among female patients remain largely unknown. Understanding the effects of sex-related genes and intestinal microbiota underlying AILD immune dysregulation, as well as the role of sex hormones in immune cells, may pave the way for novel treatment strategies for AILD.

In PBC patients, the frequent delay in diagnosis plays an important role among the male sex, leading to more advanced liver disease at PBC presentation; consequently, the risk of ACLD, liver decompensation, HCC development and mortality is higher in male than in female patients, with worse biochemical response rates. Delayed diagnosis could be partially explained by the lower incidence of PBC in male patients, leading clinicians to not consider the disease as a first choice, was well the minor presence of PBC-related symptoms. Despite its rarity, the diagnosis of PBC should be considered in men with elevated cholestatic parameters maintaining a high index of suspicion for PBC to prevent diagnostic delays.

The association between AILDs and the spectrum of EHAIDs seems to be more common in women (Table 1). Further specifically designed studies are needed about the gender impact on EHAID prevalence in patients with AILDs.

In conclusion, gender seems to have an impact on LT in ALD; however, the sex differences with respect to the prevalence, incidence, pathogenesis, risk factors, long-term survival and prognosis in this setting have not been sufficiently explored and require further investigations.

## Figures and Tables

**Figure 1 jpm-12-00925-f001:**
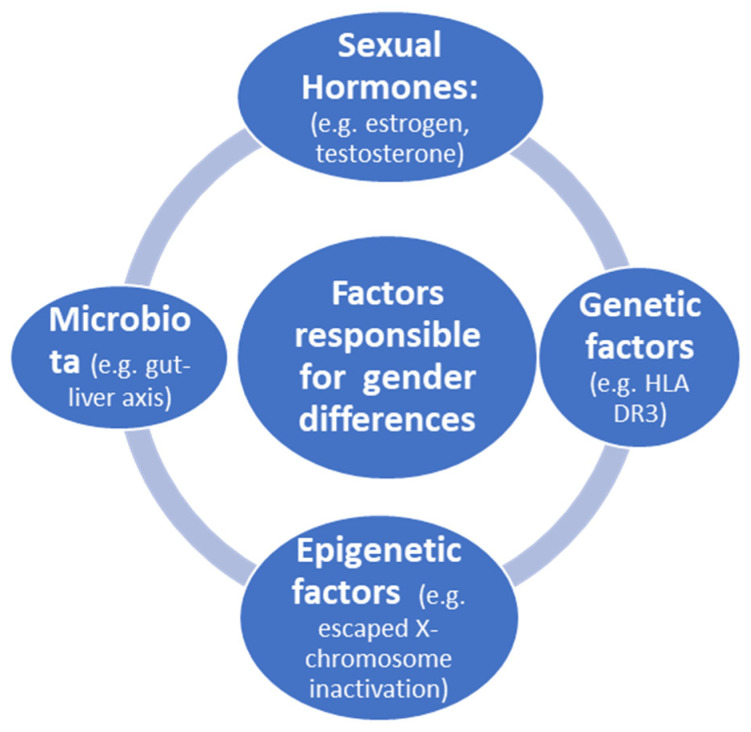
Factors responsible for gender differences in immune response.

**Table 1 jpm-12-00925-t001:** The effect of gender on the prevalence of EHAIDs in AILD.

Hepatic Autoimmune Disease	EHAID Prevalence	Most Common Association	Prevalence Distribution by Sex
AIH	30–42% (2) (3) (4)	AITDs, AISD, coeliac sprue, RA, SS, IBD (2)	90% female (3)
PBC	26.3–60% (5) (6) (7)	AITDs, SS, SSc, RA	88.6–97.3% female (6) (7)
PSC	50–80% (8)	IBD (CD)	48.1% female (9)
OS	43% (10)	AITDs	No evidence in the literature

Abbreviations: AIH: autoimmune hepatitis; PBC: primary biliary cholangitis; PSC: primary sclerosing cholangitis; OS: overlap syndrome; AITDs: autoimmune thyroid diseases (including Hashimoto thyroiditis, Graves disease and unspecified autoimmune thyroiditis); AISD: autoimmune skin disease (alopecia, vitiligo and psoriasis); RA: rheumatoid arthritis; SS: Sjogren’s syndrome; IBD: inflammatory bowel disease.

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
