# Peer review of "Gender and Autoimmune Liver Diseases: Relevant Aspects in Clinical Practice"

_jpm, 2022, doi:10.3390/jpm12060925_

Round 1
Reviewer 1 Report
This is an interesting paper on gender and autoimmune liver disease. Please comment on the characteristics of this disease regarding gender compared to autoimmune diseases of other organs.
Author Response
# Rewier 1
“Please comment on the characteristics of this disease regarding gender compared to autoimmune diseases of other organs.”
Autoimmune diseases (AD) in general are more common in females than in males. The reason for the gender difference remains unknown; however, the most convincing explanation of female biased autoimmunity remains the hormonal theory because estrogens are potent stimulators of autoimmunity and androgens seem to play a protective role in the process. Female predominance is especially characteristic in Sjogren's syndrome, systemic sclerosis, autoimmune thyroid disease and primary biliary cirrhosis. Sex differences also exist in other organ specific autoimmune diseases such as celiac disease. Nevertheless, other organ specific autoimmune diseases (i.e. Ulcerative colitis) seemingly are not characterized with an increased prevalence in females. According with this point, it’s well known that ADs are complex and multifactorial entities. Although hormone differences may have a strong influence on the predisposition of women to ADs, there is enough evidence to state that genetic factors are important as well. It also remains unknown whether the reported prevalence of autoimmune diseases in females is affected by the under-diagnosing of men
[Lleo A, Battezzati PM, Selmi C, Gershwin ME, Podda M. Is autoimmunity a matter of sex? Autoimmun Rev 2008;7:626–30].
Reviewer 2 Report
Invernizzi et al. reviewed the association of gender with autoimmune liver diseases. The manuscript is well-written and comprehensive. Here are my concerns:
1. Abstract: change LT to liver transplantation.
2. Figure 1: change genetc to genetic. It would be nice if add some keywords in each category.
3. More specific incidence is necessary throughout the manuscript. For example, the incidence of AIH is xx/1,000 person-years and xx/1,000 person-years for men and women, respectively.
4. Apart from epigenetic factors, there are gender-related sociodemographic differences, such as BMI, smoking, alcohol, etc., which are poorly described.
5. It would be nice if generate a Figure showing key gender-related factors that may contribute to the development of autoimmune diseases.
Author Response
Please see the attachment
“1. Abstract: change LT to liver transplantation”
See paper
“2. Figure 1: change genetc to genetic. It would be nic e if add some keywords in each category “
Figure 1. Factors responsible for gender differences in immune response.
“3. More specific incidence is necessary throughout the manuscript. For example, the incidence of AIH is xx/1,000 person-years and xx/1,000 person-years for men and women, respectively.”
The incidence of AIH is greater in women than men, with rates of 1.2–3.05 and 0.26–0.89 per 100000 population, respectively, in Europe.
Contemporary meta-analyses provide varied pooled incidence rates for PBC, with annual estimates across North America, Europe and the Asia-Pacific of 2.75, 1.86 and 0.84 per 100 000 population, respectively. The sex-standardised incidence of PBC is very variable, with nationwide data from South Korea showing an adjusted annual rate of 2.87 (men) and 14.12 (women). The regional data from the Northeast of England have indicated incidence rates in men and women aged 40–59 years of 0.75 and 5.40, respectively; compared with 1.51 and 12.86 in individuals aged 60–74 years.
The overall age-adjusted PSC incidence was 0.41 (95% CI 0.32 to 0.51) per 100,000 person-years. Accordin to the gender, the age-adjusted incidence rate for males was numerically higher than females (0.45 (95% CI 0.33 - 0.61) and 0.37 (95% CI 0.26 - 0.51) per 100,000 person-years, respectively).
(Trivedi PJ, Hirschfield GM. Gut 2021;70:1989–2003. doi:10.1136/gutjnl-2020-322362)
“4. Apart from epigenetic factors, there are gender-related sociodemographic differences, such as BMI, smoking, alcohol, etc., which are poorly described.”
Environmental exposure can influence the genotype, the prevalence and the risk of developing or the severity of autoimmune disease. The role of environmental factors in the development of autoimmunity has been extensively studied showing that nutrition, drugs, infections, tobacco use, ultraviolet rays exposure, as well as physical and psychological stresses could have an important role in the pathogenesis of autoimmunity.
[Shapira Y, Agmon-Levin N, Shoenfeld Y. Defining and analyzing geoepidemiology and human autoimmunity. J Autoimmun 2010;34:J168–77],
For example, Vitamin D is an environmental factor that may underlie the development of autoimmunity. It should be noted however that, within any geographical location, exposure to sunlight between men and women could vary, and this may be dependent on lifestyle or occupational choices. That said, men tend to have more unprotected sun exposure and gender-specific responses to sunlight may play a role in autoimmune disease. Another example is the role of cosmetics in PBC, greater in women. Also mercury, pesticides and solvents exposure is higher in men.
(Myers Jr. Validation of a DNA Quantitation Method on the Biomek(R) 3000. J Forensic Sci 2010;55:1570–5).
Incidence and prevalence of most ADs are not homogeneous and their frequency differs from country to country due to socioeconomic factors (e.g. medical expertise and access to medical care) and epidemiological methodology ones (e.g. community-based analysis versus hospital-based ones).
- Moroni et al. / Autoimmunity Reviews 11 (2012) A386–A392
“5. It would be nice if generate a Figure showing key gender-related factors that may contribute to the development of autoimmune diseases.”

Round 2
Reviewer 2 Report
The authors have improved the manuscript according to my concerns. I have no more comments.